# Myeloid-Specific STAT3 Deletion Aggravates Liver Fibrosis in Mice Fed a Methionine- and Choline-Deficient Diet via Upregulation of Hepatocyte-Derived Lipocalin-2

**DOI:** 10.3390/cells14191522

**Published:** 2025-09-29

**Authors:** Kyung Eun Kim, Hyun Joo Shin, Hyeong Seok An, Eun Ae Jeong, Yundong Sun, Jiwon Oh, Jiwoo Park, Jaewoong Lee, Seung-Soon Im, Gu Seob Roh

**Affiliations:** 1Department of Anatomy and Convergence Medical Science, College of Medicine, Metabolic Dysfunction Liver Disease Research Center, Institute of Medical Science, Gyeongsang National University, Jinju 52727, Republic of Korea; kke-jws@hanmail.net (K.E.K.); k4900@hanmail.net (H.J.S.); gudtjr5287@daum.net (H.S.A.); jeasky44@naver.com (E.A.J.); sundong2587@naver.com (Y.S.); tnsdksquddnj@naver.com (J.O.); parkjiwoo@gnu.ac.kr (J.P.); 2Department of Korean Medical Science, School of Korean Medicine, Pusan National University, Yangsan 50612, Republic of Korea; jwlee25@pusan.ac.kr; 3Department of Physiology, Keimyung University School of Medicine, Daegu 42601, Republic of Korea; ssim73@knu.ac.kr

**Keywords:** STAT3, lipocalin-2, inflammation, fibrosis, liver, MCD diet, mouse

## Abstract

The signal transducer and activator of transcription 3 (STAT3) in myeloid cells suppresses proinflammatory cytokine production and reduces collagen deposition. However, its role in methionine- and choline-deficient (MCD) diet-fed mice remains unclear. This study investigates the effects of myeloid-specific STAT3 deficiency on hepatic inflammation and fibrosis in MCD diet-fed mice. Myeloid-specific STAT3 knockout (mSTAT3KO) mice were fed the MCD diet for four weeks to induce metabolic dysfunction-associated steatohepatitis (MASH). MCD diet-fed mice displayed MASH-like pathological phenotypes, including hepatic steatosis, inflammation, and fibrosis. Compared with MCD diet-fed WT mice, mSTAT3KO mice fed the MCD diet exhibited reduced hepatic lipid accumulation but increased fibrosis. Notably, mSTAT3KO mice showed elevated hepatic STAT3 and lipocalin-2 (LCN2) protein levels in hepatocytes. Some proinflammatory cytokines were increased by the MCD diet in mSTAT3KO mice, which also exhibited increased hepatocyte apoptosis. Conversely, MCD diet-induced CD36, perilipin-2, acyl-CoA thioesterase 2, and 4-hydroxynonenal proteins were reduced by mSTAT3KO. Myeloid-specific STAT3 deficiency may induce a compensatory STAT3/LCN2 axis in hepatocytes, thereby exacerbating MASH progression.

## 1. Introduction

Metabolic dysfunction-associated steatohepatitis (MASH) is characterized by steatosis, inflammation, and progressive fibrosis [1,2]. With the global prevalence of obesity and metabolic syndrome on the rise, MASH has become a major public health concern due to its potential to progress to cirrhosis and hepatocellular carcinoma [3]. A well-established murine model of MASH involves feeding a methionine- and choline-deficient (MCD) diet, which induces liver histological features similar to human MASH, including steatosis, inflammation, fibrosis, and hepatocyte necrosis [4]. However, the mechanisms regulating MASH progression remain incompletely understood.

The pathogenesis of MASH is multifactorial, involving complex interactions among lipogenesis, inflammation, fibrosis, apoptosis, oxidative stress, and autophagy [5,6]. Owing to this complexity, few therapeutic agents have been approved for MASH treatment. Recent studies have highlighted the crucial role of immune cells—particularly those of myeloid lineage—in MASH development and progression [7]. Among the implicated signaling pathways, the signal transducer and activator of transcription 3 (STAT3) has emerged as a critical regulator of both immune cell activity and hepatocyte function [8]. STAT3 is a transcription factor involved in cell survival, proliferation, fibrosis, and inflammatory responses. In liver diseases, STAT3 activation in myeloid cells generally exerts anti-inflammatory effects, whereas its role in hepatocytes is more complex and context-dependent [9,10].

Lipocalin-2 (LCN2), a STAT3-regulated protein, participates in innate immunity, ferroptosis, fibrogenesis, and oxidative stress responses [11,12]. Recent evidence shows that LCN2 is a key contributor to hepatic inflammation and fibrosis during MASH progression, yet its relationship with myeloid-specific STAT3 signaling in the MCD model has not been fully elucidated [13,14].

Here, we investigate the effects of myeloid-specific STAT3 deletion on hepatic inflammation, fatty acid oxidation, oxidative stress, apoptosis, autophagic flux, and fibrosis. We also explore compensatory STAT3/LCN2-derived signaling as a potential mechanism mediating communication between hepatocytes, Kupffer cells, and hepatic stellate cells (HSCs) in MCD diet-fed mice.

## 2. Materials and Methods

### 2.1. Animals

C57BL/6J mice with a loxP-floxed STAT3 gene (STAT3^fl/fl^) and those carrying a Cre transgene driven by the distal LysM promoter (LysM-Cre^+/+^) were generously provided by Dr. Sang-Kyu Ye (Seoul National University College of Medicine, Seoul, Republic of Korea). STAT3^fl/fl^ mice were crossed with transgenic mice expressing Cre recombinase under the control of lysozyme 2 (LysMCre) promoters to generate myeloid-specific STAT3 knockout (mSTAT3KO) mice, as previously described [15]. Myeloid-specific STAT3 deletion was confirmed by PCR at 5 weeks of age (Appendix A and Appendix A). For the MCD diet-fed MASH mouse model, wild-type (WT) and mSTAT3KO mice (6 weeks old) were fed either the MCD diet or a normal diet (ND) for 4 weeks. Mice were housed under a 12 h light/dark cycle.

### 2.2. Isolation of Bone Marrow-Derived Macrophages (BMDMs)

BMDMs were isolated from the femurs and tibiae of WT and mSTAT3KO mice. Briefly, femurs and tibiae were aseptically removed and cleaned of muscle tissue. Bone marrow was flushed out using a 10 mL syringe filled with ice-cold Dulbecco’s phosphate-buffered saline (DPBS), and the cell suspension was passed through a 70 μm cell strainer to remove debris [16]. After centrifugation at 300× *g* for 5 min at 4 °C, the cell pellet was lysed in RIFA buffer containing proteases and phosphatase inhibitors for Western blot analysis.

### 2.3. Isolation of Liver Parenchymal and Non-Parenchymal Cells

Livers were perfused using a two-step collagenase perfusion method, as previously described, to isolate parenchymal (PC) and non-parenchymal (NPC) cell fractions [13,17]. Briefly, cells were obtained via in situ perfusion with EGTA, pronase (Roche Diagnostics GmbH, Mannheim, Germany), and collagenase NB4G (Nordmark Pharma GmbH, Uetersen, Germany), followed by density gradient centrifugation in DNase I (Thermo Fisher Scientific, Waltham, MA, USA) solution. Suspensions were filtered through a 70 μm cell strainer and separated by centrifugation at 50× *g* for 3 min. The pellet and supernatant were collected as the PC and NPC fractions, respectively, and transferred to new tubes. Both fractions were washed and lysed in T-PER lysis buffer (Thermo Fisher Scientific) with protease/phosphatase inhibitors for downstream analyses.

### 2.4. Body Composition Analysis

Body fat and lean mass were measured using EchoMRI (Whole-Body Composition Analyzer, Houston, TX, USA).

### 2.5. Enzyme-Linked Immunosorbent Assay (ELISA)

Mice were anesthetized with 1.25% avertin (Sigma-Aldrich, St. Louis, MO, USA). Blood was collected from the left ventricle and centrifuged, and serum LCN2 concentrations (*n* = 6–10 mice) were determined with a mouse LCN2 ELISA kit (R&D Systems, Minneapolis, MN, USA).

### 2.6. Tissue Collection and Histological Analysis

Following perfusion to remove blood, liver tissues (*n* = 3–4 mice) were harvested and immediately fixed in 4% paraformaldehyde (PFA) at 4 °C. The fixed samples were either embedded in paraffin or processed for cryosectioning, with sections cut at 5 µm thickness. For visualization of hepatic triglyceride (TG) accumulation, Nile Red staining (Sigma-Aldrich) was applied to cryosections, whereas paraffin-embedded sections were subjected to hematoxylin and eosin (H&E) staining (Abcam, Cambridge, MA, USA). Quantitative analysis of Nile Red-positive regions was performed on three sections per liver, each measuring 500 × 500 μm^2^, using ImageJ software (Version 1.52a, NIH, Bethesda, MD, USA). Imaging was conducted with an FV3000 microscope (Olympus, Tokyo, Japan). To assess fibrotic changes, Sirius Red (Sigma-Aldrich) and Masson Trichrome (MT, Polysciences Inc., Warrington, PA, USA) staining were conducted, and sections were visualized using a slide scanner (Motic, Hong Kong, China). Positive stained areas in 14 selected fields (at 100× magnification) from three animals per group were quantitatively analyzed using ImageJ (Version 1.52a).

### 2.7. Measurement of Liver Aspartate Alanine Aminotransferase (ALT) and Aminotransferase (AST) Concentrations

Using liver ALT and AST kits (Biomax, Guri-si, Republic of Korea), liver AST and ALT activities in frozen livers (*n* = 8 mice) were measured according to the manufacturer’s protocols.

### 2.8. Hepatic Triglyceride (TG) Colorimetric Assay

Liver samples from 5–6 frozen mice were homogenized and then subjected to centrifugation, after which the resulting supernatants were analyzed for TG content using a colorimetric assay kit (Cayman Chemical Company, Ann Arbor, MI, USA).

### 2.9. Hepatic Hydroxyproline Assay

Hydroxyproline concentrations were quantified in frozen liver tissues collected from eight mice and measured using a mouse hydroxyproline assay kit (Cell Biolabs, Inc., San Diego, CA, USA).

### 2.10. MASLD Activity Score Measurement

MASH activity was quantified by simply adding together the individual scores assigned to steatosis (ranging from 0 to 3), lobular inflammation (0 to 3), and hepatocyte ballooning (0–2) [18].

### 2.11. Tissue Fractionation and Western Blot Analysis

To prepare total protein extracts from frozen liver tissues (*n* = 5–8 mice), samples were homogenized in T-PER lysis buffer (Thermo Fisher Scientific) supplemented with protease and phosphatase inhibitor cocktails. For subcellular fractionation, cytoplasmic and nuclear proteins were isolated using the NE-PER Nuclear and Cytoplasmic Extraction Kit (Pierce, Rockford, IL, USA). Protein concentrations were quantified via the BCA assay (Thermo Fisher Scientific). The primary antibodies used are shown in Appendix A. β-actin and p84 were used as loading controls to normalize protein levels in total and nuclear fractions, respectively. Protein bands were detected using enhanced chemiluminescence substrates (Thermo Fisher Scientific), and chemiluminescence was analyzed using a LAS-4000 instrument (Fujifilm, Tokyo, Japan). The Multi-Gauge V 3.0 image analysis program (Fujifilm) was used for densitometry analysis.

### 2.12. Cytokine and Chemokine Array

Pooled liver lysates obtained from four mice per group (WT or mSTAT3KO) subjected to either an ND or MCD diet were analyzed using the Proteome Profiler Mouse Cytokine Array Kit (R&D Systems) to assess the expression of 40 cytokines and chemokines following the manufacturer’s instructions.

### 2.13. RNA Isolation and Quantitative Real-Time PCR (RT-PCR)

Total RNA was isolated from liver tissues (*n* = 5–6 mice) using TRIzol reagent (Invitrogen, Waltham, MA, USA) and converted to cDNA with the RevertAid First Strand cDNA Synthesis Kit (Thermo Fisher Scientific). Quantitative RT-PCR was conducted on a LightCycler 480 Instrument II (Roche Diagnostics GmbH) using iQ™ SYBR Green Supermix (Bio-Rad Laboratories, Hercules, CA, USA) and the primer sets listed in Appendix A. Relative transcript levels were determined by using the ∆∆Ct method, expressed as fold-change compared with a designated calibrator sample. Each reaction was performed in duplicate, and mean values were used for analysis.

### 2.14. Terminal Deoxynucleotidyl Transferase dUTP Nick End Labeling (TUNEL) Assay

The proportion of GS-positive apoptotic adipocytes was assessed using the In-Situ Cell Death Detection Kit (Roche Molecular Diagnostics, Mannheim, Germany) following the manufacturer’s instructions. Nuclei were counterstained with 4′,6-diamidino-2-phenylindole (DAPI, Invitrogen), and the sections were visualized on an FV3000 microscope (Olympus). TUNEL-positive regions were quantified from 15 fields (×100 magnification) obtained from three mice per group using ImageJ software (Version 152a).

### 2.15. Immunohistochemistry

Deparaffinized liver sections were treated with 0.3% H_2_O_2_ for 30 min, rinsed, and subsequently incubated with normal serum for 1 h to block nonspecific binding. Sections were then incubated overnight at 4 °C with primary antibodies (Appendix A), followed by a 1 h incubation with biotinylated secondary antibodies. After additional washes, an avidin–biotin–peroxidase complex (Vector Laboratories, Burlingame, CA, USA) was applied, and staining was developed using a 0.05% diaminobenzidine substrate kit (Vector Laboratories). The slides were imaged using a microscope slide scanner (Motic).

### 2.16. Transmission Electron Microscopy (TEM)

Mice (*n* = 2 mice) were anesthetized and perfused with 2% PFA and 2% glutaraldehyde in 0.1 M PBS. Liver samples were further fixed in the same solution for 18 h at 4 °C, followed by rinsing in 0.1 M PBS and post-fixation in 1% osmium tetroxide for 90 min. Tissue sections were dehydrated through an ethanol gradient, treated with propylene oxide for 10 min, and embedded using the Poly/Bed 812 Embedding Kit (Polysciences Inc., Warrington, FL, USA) for 18 h. Ultrathin sections (60 nm) were prepared with a diamond knife, then stained with 5% uranyl acetate for 10 min and 1% lead citrate for 5 min. Images were acquired using a JEM-1011 transmission electron microscope (JEOL Ltd., Tokyo, Japan) operated at 80 kV and recorded digitally (EMSIS GmbH, Muenster, Germany).

### 2.17. Statistical Analysis

Statistical analysis was carried out using PRISM version 10.0 (GraphPad Software Inc., San Diego, CA, USA). Differences between groups were assessed by using two-way ANOVA, followed by Bonferroni’s post hoc testing. The results are presented as the mean ± standard error of the mean (SEM), with statistical significance defined as *p* < 0.05.

## 3. Results

### 3.1. Generation of Myeloid-Specific STAT3-Deleted Mice

To explore the role of myeloid STAT3 in MCD diet-induced MASH, STAT3^fl/fl^ mice were crossed with transgenic mice expressing Cre recombinase under the control of lysozyme 2 (LysMCre) promoters to generate mSTAT3KO mice (Appendix A). Although there was no change in STAT3 expression in various tissues, including liver, epididymal fat pads, spleen, brain, lung, and skeletal muscle, between WT and mSTAT3KO mice (Appendix A), the STAT3 protein from BMDM was absent in mSTAT3KO mice compared to STAT3^fl/fl^ (WT) mice (Appendix A). Both phosphorylated (p)-STAT3 and STAT3 protein expressions were partially deleted in isolated NPC, including Kupffer cells (Appendix A). However, their protein expressions were dramatically increased in isolated PC (hepatocytes) from mSTAT3KO mice compared to WT mice (Appendix A).

### 3.2. Myeloid-Specific STAT3 Deletion Reduces Intrahepatic Lipid Accumulation in MCD Diet-Fed Mice

Mice fed the MCD diet for 4 weeks showed a significant reduction in weight, including body, fat mass, lean mass, and liver (Figure 1A–D). However, these weight changes were not affected by mSTAT3KO. Liver ALT activity was significantly increased by mSTAT3KO under the MCD diet, whereas AST activity was unaffected (Figure 1E,F). MCD diet-fed WT mice exhibited histological changes characterized by hepatic steatosis, inflammation, and ballooning hepatocytes (Figure 1G,H). Furthermore, Nile Red staining and TEM revealed that mSTAT3KO significantly attenuated many large lipid droplets (LDs) within hepatocytes in MCD diet-fed WT mice (Figure 1G). In accordance with histological analyses, increased hepatic TG levels in MCD diet-fed WT mice were attenuated by mSTAT3KO (Figure 1I). Given that the MCD diet induces intrahepatic lipid accumulation by reducing very low-density lipoprotein secretion [19], we determined de novo lipogenesis-related proteins. As expected, we found that the MCD diet causes reduced nuclear mature sterol regulatory element binding protein-1c (mSRBEP-1c), acetyl-CoA carboxylase (ACC), fatty acid synthase (FAS), and stearoyl-CoA desaturase1 (SCD1) protein expressions in the liver compared to the ND diet (Appendix A). However, these de novo lipogenesis-related protein expressions were not affected by mSTAT3KO. These findings indicate that myeloid-specific STAT3 deletion can attenuate intrahepatic lipid accumulation in MCD diet-fed mice with independent de novo lipogenesis.

### 3.3. Myeloid-Specific STAT3 Deletion Promotes Hepatic Fibrosis in MCD Diet-Fed Mice

In addition to intrahepatic lipid accumulation, the MCD diet itself induces advanced perisinusoidal and periportal fibrosis [19]. To understand the potential mechanisms by which mSTAT3KO induces hepatic fibrotic changes in MCD diet-fed mice, we performed histological staining and quantified fibrogenic proteins. Sirius Red and Masson Trichrome staining revealed that MCD diet-induced perisinusoidal fibrosis in WT mice is enhanced by mSTAT3KO (Figure 2A,B). In addition, we found that mSTAT3KO increases MCD diet-induced hydroxyproline levels in liver tissues compared to MCD diet-fed WT mice (Figure 2C). Some fibrogenic proteins, such as matrix metalloproteinase 9 (MMP9), lumican, and α-smooth muscle actin (α-SMA), were significantly increased by the MCD diet or mSTAT3KO (Figure 2D). Unexpectedly, the MCD diet aggravated mSTAT3KO-induced MMP9, lumican, and α-SMA protein levels compared to MCD diet-fed WT mice. Our results indicate that although myeloid-specific STAT3 deletion attenuates intrahepatic lipid accumulation, it paradoxically promotes MCD diet-induced hepatic fibrosis.

### 3.4. Myeloid STAT3 Deletion Increases Hepatic STAT3 and LCN2 Proteins

As shown in Appendix A, modest STAT3 activation occurred in MCD diet-fed WT mice; however, total and phosphorylated STAT3 levels were markedly higher in both ND diet-fed and MCD diet-fed mSTAT3KO mice compared with MCD diet-fed WT mice (Figure 3A). Next, we determined the effects of mSTAT3KO on hepatic LCN2, a transcriptional target of STAT3. Similar to the hepatic STAT3 pattern, circulating LCN2 and hepatic *lcn2* mRNA levels were elevated in mSTAT3KO mice, especially under MCD feeding (Figure 3B,C). However, mSTAT3KO promoted MCD diet-induced LCN2 protein in the livers of MCD diet-fed mice (Figure 3D). Immunohistochemistry revealed that while LCN2-positive cells are predominantly in Kupffer cells in MCD diet-fed WT and ND diet-fed mSTAT3KO mice, MCD diet-fed mSTAT3KO mice displayed strong LCN2 staining in both hepatocytes and Kupffer cells (Figure 3E). PC and NPC fractionation confirmed that in addition to the increases in LCN2 in only NPCs of MCD diet-fed WT mice, mSTAT3KO causes a prominent increase in its protein from both PC and NPC fractions (Figure 3F), suggesting a compensatory increase in hepatocyte-derived LCN2 upregulation in response to myeloid STAT3 loss.

### 3.5. mSTAT3KO Mice Are More Susceptible to MCD Diet-Induced Hepatic Inflammation

Given that STAT3/LCN2 signaling-related fibrosis was linked to hepatic inflammation, we determined NF-kBp65-related inflammatory cytokine and chemokines using Western blot analysis and a proteome profiler cytokine array. Not only the MCD diet, but also mSTAT3KO itself caused a significant inflammatory response—related NF-kBp65, interleukin-6 (IL-6), and high-mobility group box1 (HMGB1) proteins were increased, and they were enhanced in MCD diet-fed mSTAT3KO mice (Figure 4A). However, mSTAT3KO-induced galectin-3 was not promoted by the MCD diet. Using a liver cytokine array, we found that compared to ND diet-fed WT mice, C-X-C motif chemokine 13 (CXCLl3), C5a, tissue inhibitor matrix metalloproteinase1 (TIMP1), triggering receptor expressed on myeloid cells1 (TREM1), CD54, CXCL9, IL-1α, IL-16, IL-1ra, and chemokine ligand 5 (CCL5) were increased in MCD diet-fed WT or ND diet-fed mSTAT3KO mice (Figure 4B). However, the increased levels of C5a and CXCL12 in ND diet-fed mSTAT3KO mice were decreased by mSTAT3KO. These findings imply that mSTAT3KO could promote MCD diet-induced hepatic inflammation, resulting in upregulation of hepatic fibrosis.

### 3.6. mSTAT3KO Mice Are More Susceptible to MCD Diet-Induced Hepatocyte Apoptosis

To assess whether mSTAT3KO promotes liver apoptosis and regeneration in MCD diet-fed mice, we evaluated the effects of the MCD diet on hepatic glutamine synthetase (GS), a susceptible protein to oxidative damage. Compared with MCD diet-fed WT mice, hepatic GS protein was reduced in MCD diet-fed STAT3KO mice (Figure 5A). TUNEL staining also showed a dramatic increase in TUNEL-positive cells in MCD diet-fed mSTAT3KO mice compared with MCD diet-fed WT mice (Figure 5B). To determine whether the MCD diet affects liver regeneration in response to hepatocyte apoptosis, we assessed p21, proliferating cell nuclear antigen (PCNA), and Ki67 protein levels (Figure 5C,D). As expected, these proteins were significantly increased by only the MCD diet. Although mSTAT3KO caused a slight increase in p21 and PCNA protein levels, they were not promoted by mSTAT3KO (Figure 5C). Immunohistochemistry revealed that many Ki67-positive hepatocytes were observed in both MCD diet-fed WT and mSTAT3KO mice (Figure 5D). These data suggest that myeloid STAT3 may play a protective role in hepatocyte apoptosis in MCD diet-induced MASH.

### 3.7. Myeloid-Specific STAT3 Deletion Decreases Hepatic Peroxisome Activity in MCD Diet-Fed Mice

LCN2 is an essential mediator linking hepatocyte apoptosis and peroxisome activity with intracellular fatty acid (FA) levels [20]. In particular, peroxisomal function is involved in oxidative reactions, including the oxidation of very-long-chain fatty acids. To assess peroxisomal activity, we determined peroxisomal membrane protein 11A (PEX11A) and catalase (classical marker protein of peroxisome) protein levels. These protein levels were reduced in MCD diet-fed and mSTAT3 mice compared with ND diet-fed WT mice (Figure 6A). Immunohistochemistry also revealed that compared to ND diet-fed WT mice, less immuno-stained PEX11A-positive hepatocytes were observed in MCD diet-fed and mSTAT3KO mice (Figure 6B). TEM revealed that in ND diet-fed WT mice, close contacts between large LD and peroxisome were observed, while these contacts disappeared in the hepatocytes of MCD diet-fed and mSTAT3KO mice (Figure 6C). So, our findings indicate that hepatocyte-derived LCN2 may be linked to reduced peroxisomal activity and mitochondrial FA oxidation induced by the MCD diet.

### 3.8. Myeloid-Specific STAT3 Deletion Reduces MCD Diet-Induced Mitochondrial FA Oxidation and Oxidative Stress

Although mSTAT3KO caused a significant reduction in hepatic TG contents in MCD diet-fed mice, MCD diet-fed mSTAT3KO mice showed aggravated hepatic inflammation, fibrosis, and apoptosis. To understand these contrasting phenotypes, we determined FA oxidation and oxidative stress-related protein expression using Western blot analysis. Under MCD diet-induced CD36 protein conditions, hepatic peroxisome proliferator-activated receptor α (PPARα), the principal transcriptional factor governing FA oxidation, was reduced by the MCD diet and mSTAT3KO (Figure 7A). However, increased hepatic CD36 protein in MCD diet-fed WT mice was significantly reduced by mSTAT3KO. Mitochondrial FA oxidation plays an essential role in the regulation of intrahepatic lipid accumulation during intracellular lipolysis for energy production. Following lipolysis, long-chain FAs are transported into the mitochondria [21]. Among the Acyl-CoA Thioesterase (ACOT) family, ACOT2 localizes to the mitochondrial matrix and hydrolyses long-chain fatty acyl-CoA [22]. In line with the altered expression of PEX11A and catalase protein (Figure 6A), we notably found that hepatic ACOT2 was prominently reduced by mSTAT3KO (Figure 7A). However, mSTAT3KO significantly reduced MCD diet-induced ACOT2 protein levels compared to those in MCD diet-fed WT mice.

We next determined the antioxidant enzymes against the MCD diet with or without mSTAT3KO. We found that heme oxygenase-1 (HO-1) and NAD (P) H: quinone oxidoreductase-1 (NQO-1) protein levels were increased in MCD diet-fed WT mice compared with ND diet-fed WT mice (Figure 7B). However, these increased antioxidant proteins were not promoted by mSTAT3KO. In addition, the MCD diet significantly increased hepatic glutathione peroxidase-4 (GPX-4) and 4-hydroxynonenal (4-HNE) protein levels. In contrast, these proteins were reduced by mSTAT3KO (Figure 7B). Taken together, these findings suggest that myeloid STAT3 may play an important role in peroxisome activity-related mitochondrial FA oxidation and oxidative stress in MCD diet-induced MASH.

### 3.9. Myeloid-Specific STAT3 Deletion Impairs Hepatic Autophagic Flux in MCD Diet-Fed Mice

Given that microtubule-associated protein 1A/1B–light chain 3 (LC3) lipidation and its association with autophagosome membranes have been demonstrated to be valuable autophagy markers, we investigated autophagic flux using LC3B and p62 antibodies. As expected, a significant increase in p62 protein was shown in MCD diet-fed WT mice compared to ND diet-fed WT mice (Figure 8A). After autophagosomes form, they fuse with lysosomes for hydrolytic digestion [23]. So, we examined whether the MCD diet alters transcription factor EB (TFEB) protein levels, the master regulator of lysosomal gene expression. Compared to ND diet-fed WT mice, total and cytosolic TFEB protein levels were significantly increased in the livers of MCD diet-fed WT mice (Figure 8B). However, nuclear TFEB protein was prominently reduced in MCD diet-fed WT and mSTAT3KO mice. We next found that lysosomal-associated membrane protein (LAMP1), a marker of lysosome, was significantly increased in MCD diet-fed mice, while its protein was reduced in MCD diet-fed mSTAT3KO mice (Figure 8B). TEM revealed that large LDs and lipolysosomes (Ly) within hepatocytes were observed in MCD diet-fed WT mice compared to ND diet-fed WT mice (Figure 8C). These findings indicate that mSTAT3 could partly impair autophagic flux within hepatocytes in MCD diet-fed mice.

## 4. Discussion

Our findings provide new insight into the complex interplay between myeloid cells and hepatocytes in the development of a MASH-like liver phenotype with advanced fibrosis. An unexpected observation was that myeloid-specific STAT3 deletion exacerbates liver fibrosis by upregulating hepatocyte-derived LCN2, which in turn can activate HSCs.

We propose that the observed decrease in hepatic peroxisomal activity in mSTAT3KO mice may lead to reduced mitochondrial FA oxidation and thereby accelerate the progression of MCD diet-induced MASH.

Compared with Western diet models, the MCD diet rapidly reproduces human MASH etiology and leads to advanced fibrosis [24,25]. In general, the MCD diet induces early steatosis and inflammation within 2–4 weeks, while robust fibrosis is more prevalent after a minimum of 6–8 weeks of the diet [26]. We found that after 4 weeks of implementing the MCD diet, mice displayed perisinusoidal fibrosis and increased profibrotic markers. However, the MCD model should be interpreted with caution because of the significant weight loss. Consistent with MCD diet-induced weight loss in db/db mice [4], we showed approximately a 50% loss of body weight, including decreases in liver, fat, and lean mass after four weeks of MCD feeding. This weight loss is thought to result from hypermetabolism associated with suppressed hepatic SCD1 activity [27]. In our study, mSTAT3KO did not alter hepatic expression of de novo lipogenesis-related proteins, nor did it influence MCD diet-induced weight loss, suggesting that myeloid-specific STAT3 deletion does not affect MCD diet-induced systemic hypermetabolism.

STAT3 can have both anti- and proinflammatory effects in liver fibrosis, depending on the cell type in which it is activated [8,28]. In hepatocytes, STAT3 plays a critical role in inhibiting steatosis, whereas in macrophages, its activation suppresses liver inflammation during chronic alcohol liver injury [29]. In other disease models, mSTAT3KO increased macrophage activation and enhanced MMP9 expression during Staphylococcal pneumonia [30]. Anther study reported that deletion of STAT3 in myeloid cells caused increased infiltration of macrophages and neutrophils into the remnant liver following partial hepatectomy [10]. However, the novel aspect of the current study is the demonstration that myeloid STAT3 deficiency not only increases hepatic inflammation but also triggers compensatory hepatocyte STAT3 activation, dramatically elevating LCN2 levels—uncovering a previously unrecognized STAT3/LCN2 axis driving fibrogenesis, even when hepatic lipid accumulation is attenuated. Furthermore, STAT3 activation is responsible for creating an inflammatory microenvironment that facilitates lipid accumulation, leading to the dysregulation of lipid metabolism genes [31]. Thus, mSTAT3KO mice are a valuable model to dissect the specific contributions of STAT3 in myeloid cells versus hepatocytes [32]. Based on this study, our results demonstrate that the activation of STAT3 in mSTAT3KO mice paradoxically drives hepatocyte STAT3 activation and enhances LCN2/MMP9 signaling, which are associated with fibrogenesis. As shown in Appendix A–E, although STAT3 protein was reduced in BMDM and NPC lysates, STAT3 was increased in isolated hepatocytes. In mSTAT3KO mice, pSTAT3 protein was also significantly increased, while the MCD diet slightly decreased the activation of STAT3. As shown in Figure 3A, STAT3 activation promoted hepatic inflammation and fibrosis in MCD diet-fed mSTAT3KO mice, suggesting that mSTAT3KO-induced hepatic STAT3 activation may play a detrimental role in MCD diet-induced MASH. So, we propose the following molecular mechanisms for the compensatory effects of myeloid STAT3 deficiency: First, the inflammatory response from myeloid STAT3 loss leads to upregulated NF-kBp65-mediated IL-6 and other cytokines, which bind to their receptors on hepatocytes and potently activate JAK-dependent phosphorylation of STAT3 in these cells. Second, hepatocytes increase STAT3 and LCN2 expression as an adaptive, compensatory measure to counterbalance the immune imbalance and promote cell survival, but this also activates downstream fibrogenic programs. This enhanced STAT3/LCN2 axis signaling in hepatocytes in mSTAT3KO mice contributes to HSC activation and fibrosis, representing a maladaptive paracrine loop driven by immune cell dysfunction. So, to understand the mechanism by which mSTAT3KO mice are susceptible to MCD-induced MASH, future studies should be conducted using hepatocyte-specific STAT3-deficient mice.

We notably found that in line with the altered expression of hepatic STAT3 protein, hepatic LCN2 proteins in mSTAT3KO mice were dramatically increased compared with MCD diet-fed WT mice. The MCD diet prominently increased hepatic LCN2 and MMP9 expression in mSTAT3KO mice. In an activated immune condition of mSTAT3KO mice, as well as strong LCN2-exhibiting Kupffer cells, many LCN2-expressing hepatocytes could play an important role in lipid accumulation, inflammation, fibrosis, and apoptosis. In contrast to our findings, MCD diet-fed LCN2-deficient mice develop prominent steatosis that is characterized by elevated levels of intrahepatic LDs [33]. In MCD diet-fed mSTAT3KO mice, we found that increased LCN2-expressing hepatocytes have less intrahepatic LD and more apoptosis. However, these mice developed a MASH-like liver phenotype with advanced fibrosis. Thus, we strongly suggest that hepatocyte-derived LCN2 from mSTAT3KO mice could promote inflammation and fibrosis after adopting an MCD diet. These findings support our previous study that HSC activation-related fibrosis is cross-linked with the upregulation of LCN2 and MMP9 in HFD-fed ob/ob mice [13]. Another group reported that chronic treatment with recombinant LCN2 exacerbated diet-induced MASH through the induction of a chemokine (C-X-C motif) receptor [34]. As shown in Figure 4B, the cytokine and chemokine array revealed increased expression of TIMP1 (tissue inhibitor of metalloproteinase), CCL5, HMGB1, and CXCL13, all of which contribute to a fibrogenic microenvironment and extracellular matrix remodeling. Thus, this upregulation of hepatocyte LCN2, coupled with enhanced STAT3 activation in hepatocytes, suggests a novel pathway by which hepatocytes adapt to alterations in myeloid cell function. The proinflammatory effects of LCN2 for MASH progression, including increased STAT3 and NF-κBp65, highlight its potential as an anti-inflammatory target.

In addition to the inflammatory response, we showed that an MCD diet increases both hepatocyte regeneration and apoptosis, and that mSTAT3KO accelerates liver apoptosis, with significantly higher levels of TUNEL-positive cells. In accordance with a study showing that an MCD diet accelerates hepatocyte proliferation and liver regeneration, particularly after partial hepatectomy [35], our findings indicate that cell-specific STAT3 may play an important role in protection against MCD diet-induced hepatocyte damage.

An MCD diet induces significant changes in the expression of genes related to lipid metabolism and oxidative stress in hepatocytes, with another study reporting downregulation of genes involved in FA oxidation and peroxisomal activity [36]. A previous study showed that PEX11A deficiency reduces the number of peroxisomes in brown adipose tissue, which is related to impaired FA oxidation and the accumulation of very-long-chain fatty acids [37]. Consistent with a study showing that reduced PPARα expression leads to decreased transcription of genes required for peroxisomal function, impairing FA metabolism [38], we also found that an MCD diet attenuated hepatic PEX11A, catalase, and PPARα protein expression. Although the MCD diet reduced hepatic PEX11A and catalase proteins, antioxidant enzymes, such as HO-1, NQO-1, and GPX-4, were inversely increased by the MCD diet. These data imply that impaired peroxisomal activity and oxidative stress induced by the MCD diet contribute to LD accumulation within hepatocytes. However, we showed that although mSTAT3KO reduced these peroxisomal activity-related proteins, MCD diet-induced GPX-4 and 4-HNE protein levels are not increased by mSTAT3KO. Therefore, these data suggest that downregulated peroxisomal activity caused by mSTAT3KO may contribute to MCD diet-induced inflammation and fibrosis.

Consistent with evidence that an MCD diet increases FA uptake [4], our findings show that hepatic CD36 protein is increased by an MCD diet. As shown in Figure 1G and I, hepatic lipid accumulation and TG concentration are increased in MCD diet-fed WT mice. In steatotic hepatocytes, lipid metabolism is regulated by complex crosstalk among the FA transporter CD36, nuclear PPARα, and mitochondrial enzyme ACOT2. We suggest that their interplay can influence FA uptake, oxidation, and storage, shaping the pathogenesis of hepatic steatosis. In MCD diet-fed WT mice, hepatic CD36 and ACOT2 protein were increased, but PPARα was reduced, suggesting that although loss of PPARα activity impairs FA oxidation, ACOT2 for mitochondrial β-oxidation facilitates the adaptation to increased FA flux. These positive effects were enhanced by mSTAT3KO. Of note, we found that hepatic ACOT2 is significantly reduced in ND diet-fed mSTAT3KO mice compared to ND diet-fed WT mice. Consistent with our findings, interference with the *Acot2* gene markedly inhibited LD accumulation and TG content in adipocytes, while its overexpression promoted both [39]. So, our data show that hepatocyte ACOT2 may play an important role in mitochondrial FA oxidation in response to hepatic LC accumulation.

On the other hand, although SREBP-1c transcriptionally regulates de novo lipogenesis-related genes, both FAS and SCD1 can be downregulated by other factors, such as leptin [40]. However, MCD feeding suppresses de novo lipogenesis through decreasing FAS and SCD1 gene expression [41]. Although SREBP-1c was slightly reduced by the MCD diet, hepatic ACC, FAS, and SCD1 protein expressions were significantly reduced in both MCD diet-fed WT and mSTAT3KO mice. We found that hepatic ACC, FAS, and SCD1 protein levels were also not changed by mSTAT3KO. These findings indicate that although myeloid-specific STAT3 deficiency has no effect on de novo lipogenesis in MCD diet conditions, it can affect the reduction in intrahepatic lipid accumulation by downregulating CD36 and ACOT2.

This study revealed that although mSTAT3KO aggravates MCD diet-induced hepatic fibrosis, it reduces intrahepatic lipid accumulation in MCD diet-fed mice. So, we suggest that an increase in STAT3/LCN2-related signaling in hepatocytes may be linked to selective autophagy that targets lipid droplets for degradation. Autophagy is a catabolic process that targets and transports cellular proteins and damaged organelles to lysosomes for degradation to promote cell survival and maintain cellular homeostasis [42]. In particular, lipophagy is involved in the clearance of LDs and TGs [43,44]. So, lipophagy impairment is associated with the progression of MASLD [45], which is in accordance with a study the accumulation of p62 in patients with MASH [46]. The MCD diet decreased LC3II and increased hepatic p62 proteins in WT and mSTAT3KO mice, indicating impairment of the autophagic flux. However, a recent study has reported that p62 was increased, but LAMP1 and TFEB were reduced in MCD diet-fed mice and MCD mimic LO2 cells [47]. They found that ponatinib, an active tyrosine kinase inhibitor, ameliorated MCD diet-induced MASH by enhancing autophagy through increased TFEB expression. Furthermore, TEM revealed that lipolysosomes were observed within the hepatocytes of MCD diet-fed mice, indicating upregulation of hepatic LAMP1 and TFEB protein expression. These data suggest that TFEB, as a major regulator of lysosome, has the ability to regulate lipophagy by promoting the production and function of lysosomes against impairment of the autophagic flux induced by an MCD diet. In MCD diet-fed mSTAT3KO mice, LC3BII, p62, and TFEB proteins were increased, but only LAMP1 was decreased compared to in MCD diet-fed WT mice. We suggest that as increased hepatic CD36 and LDs are attenuated by mSTAT3KO, LAMP1 is not promoted in MCD diet-fed mSTAT3KO mice. Taken together, our findings indicate that mSTAT3KO not only increases inflammation, but also probably promotes fibrosis through impairment of the autophagic flux in the livers of MCD diet-fed mice.

## 5. Conclusions

Myeloid-specific STAT3 deletion in the MCD model results in a complex phenotype—attenuated steatosis but aggravated inflammation, apoptosis, and fibrosis—through the activation of a hepatocyte STAT3/LCN2 axis, suppression of peroxisome and mitochondrial β-oxidation, and impairment of lipophagy. In particular, our findings suggest that mSTAT3KO has important clinical implications in MASH, highlighting both the risks of dysregulated inflammation and the therapeutic potential of cell-targeted modulation. Myeloid-specific activation of STAT3 supports anti-inflammatory effects and limits fibrosis, whereas excessive hepatocyte STAT3 activation, especially when compensatory, can drive fibrogenesis. Pharmacologic STAT3 inhibition in HSCs can reduce fibrosis by suppressing profibrotic gene expression, suggesting the need for selective, cell type-specific interventions. In addition, circulating levels of LCN2 can serve as noninvasive biomarkers to monitor disease progression and therapeutic response in MASH patients. These effects highlight the critical role of intercellular STAT3 signaling in MASH progression and underscore the need for future studies using hepatocyte-specific STAT3 knockouts to disentangle the cell type-specific functions of this pathway.

## Figures and Tables

**Figure 1 cells-14-01522-f001:**
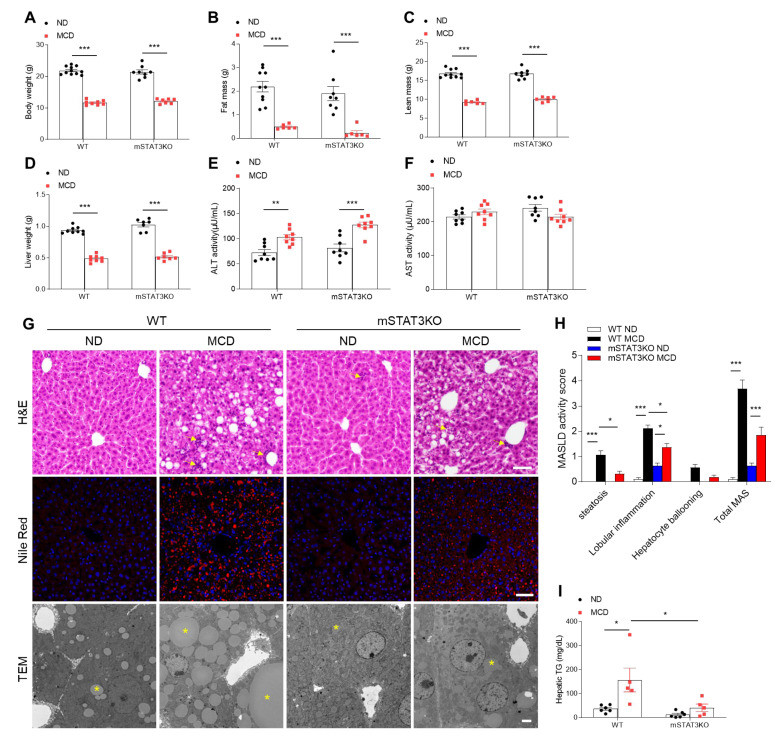
Effect of myeloid-specific STAT3 deletion on hepatic lipid accumulation in MCD diet-fed mice. WT and mSTAT3KO mice were fed an ND or MCD diet for 4 weeks to induce MASH. (**A**) Body weight (*n* = 7–11 mice per group). (**B**) Fat mass (*n* = 6–10 mice per group). (**C**) Lean mass (*n* = 6–10 mice per group). (**D**) Liver weight (*n* = 7–9 mice per group). (**E**,**F**) Hepatic ALT and AST activity (*n* = 8 mice per group). (**G**) Representative images of hematoxylin and eosin (H&E) staining, Nile red staining of liver sections (scale bar, 50 μm), and transmission electron micrographs of hepatocytes (scale bar, 2 μm). Yellow arrows and asterisks indicate neutrophils and lipid droplets (LDs), respectively. (**H**) MASLD activity score. (**I**) Hepatic triglyceride (TG) levels (*n* = 5–6 mice per group). Statistical significance was determined by using two-way ANOVA with Bonferroni’s post hoc testing. * *p* < 0.05, ** *p* < 0.001, and *** *p* < 0.0001.

**Figure 2 cells-14-01522-f002:**
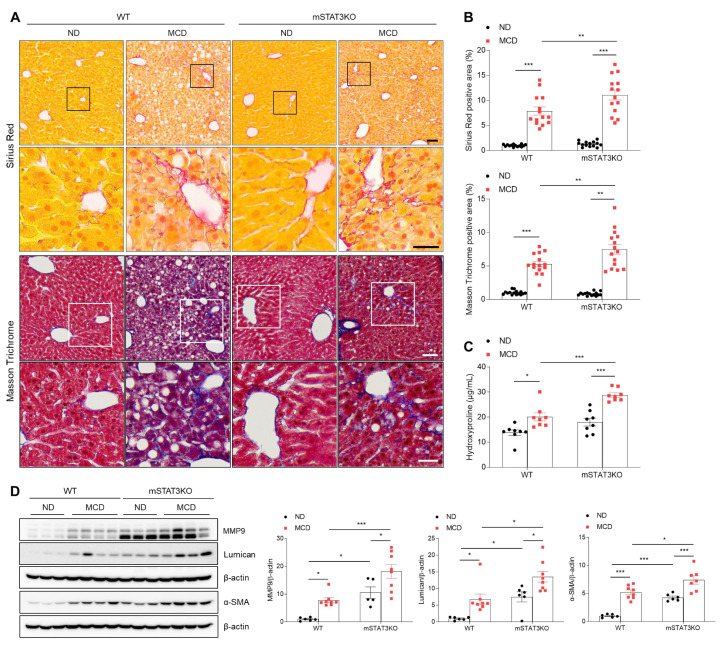
The effect of myeloid-specific STAT3 deletion on hepatic fibrosis in MCD diet-fed mice. (**A**) Representative images of Sirius Red and Masson Trichome staining of liver sections. The lower panels in each staining indicate magnified images of black and white line boxes. Scale bars: 60 μm (upper panel) and 30 μm (lower panel). (**B**) Quantification of Sirius Red- and Masson Trichome-positive areas. (**C**) Liver hydroxyproline levels from liver tissues. (**D**) Western blot analysis and quantification of MMP9, lumican, and α-SMA proteins in liver lysates (*n* = 5–8 mice per group). β-actin was used as a loading control. Statistical significance was determined by using two-way ANOVA with Bonferroni’s post hoc testing. * *p* < 0.05, ** *p* < 0.001, and *** *p* < 0.0001.

**Figure 3 cells-14-01522-f003:**
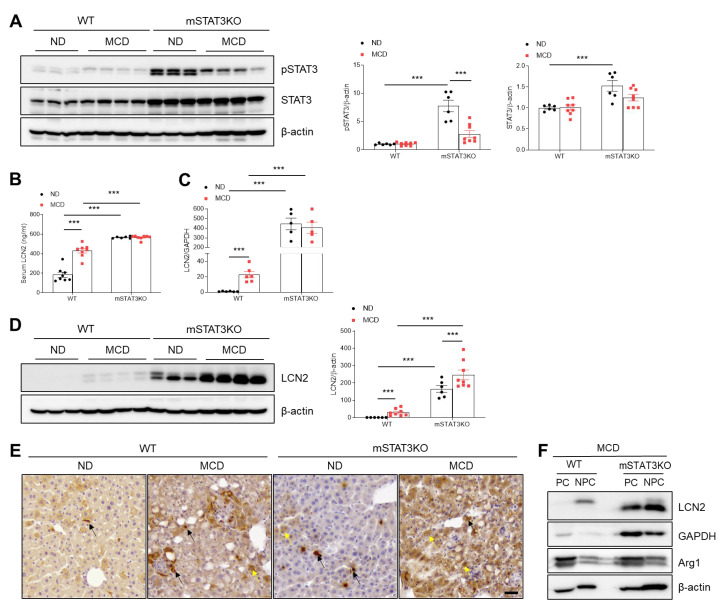
The effect of myeloid-specific STAT3 deletion on hepatic pSTAT3 and LCN2 protein expression in MCD diet-fed mice. (**A**) Western blot analysis and quantification of pSTAT3 and STAT3 proteins in liver lysates (*n* = 6–8 mice per group). β-actin was used as a loading control. (**B**) Serum LCN2 levels (*n* = 6–8). (**C**) LCN2 mRNA levels in liver tissue (*n* = 5–6 mice per group). (**D**) Western blot analysis and quantification of LCN2 proteins in liver lysates (*n* = 6–8 mice per group). β-actin was used as a loading control. (**E**) Immunohistochemistry for LCN2 in liver sections. Black and yellow arrows indicate Kupffer cells and hepatocytes, respectively. Scale bar: 30 μm. (**F**) Western blot analysis of LCN2 protein in PCs and NPCs isolated from the livers of MCD diet-fed WT and mSTAT3KO mice. GAPDH and β-actin were used as loading controls and Arg1 served as a hepatocyte marker. Statistical significance was determined by using two-way ANOVA with Bonferroni’s post hoc testing. Statistical significance between WT ND and WT MCD in (**C**,**D**) was assessed using an unpaired two-tailed Student’s *t*-test. *** *p* < 0.0001.

**Figure 4 cells-14-01522-f004:**
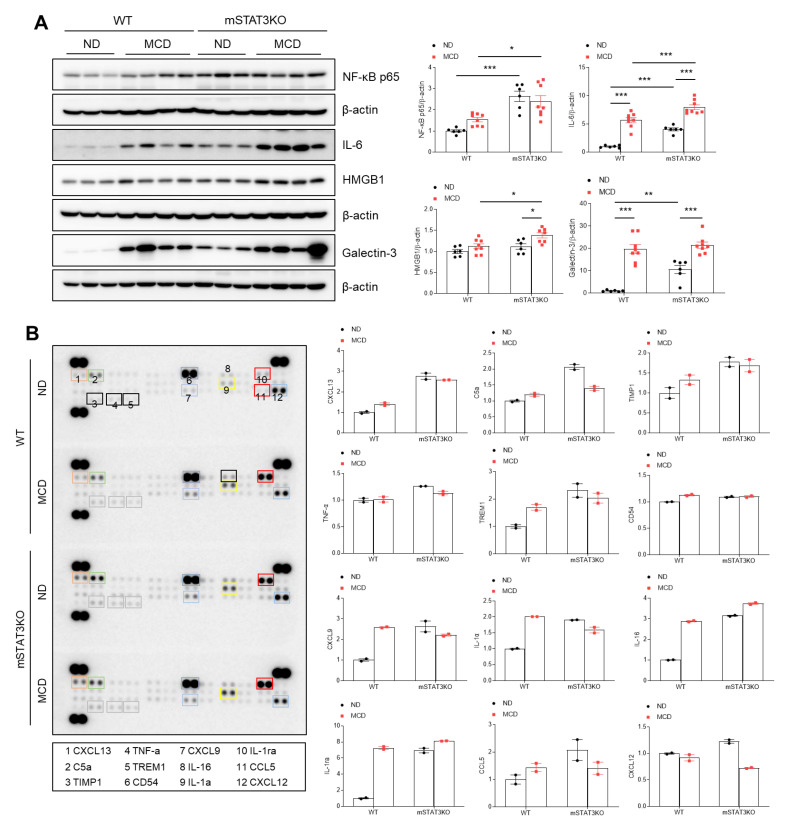
The effect of myeloid-specific STAT3 deletion on hepatic inflammation in MCD diet-fed mice. (**A**) Western blot analysis and quantification of NF-κB p65, IL-6, HMGB1, and Galectin-3 proteins in liver lysates (*n* = 6–8 mice per group). β-actin was used as a loading control. (**B**) Expression of cytokines and chemokines in the liver lysates (*n* = 4 mice per group) was analyzed using cytokine arrays. The bar graph shows the mean pixel density for each cytokine. Statistical significance was determined by using two-way ANOVA with Bonferroni’s post hoc testing. * *p* < 0.05, ** *p* < 0.001, and *** *p* < 0.0001.

**Figure 5 cells-14-01522-f005:**
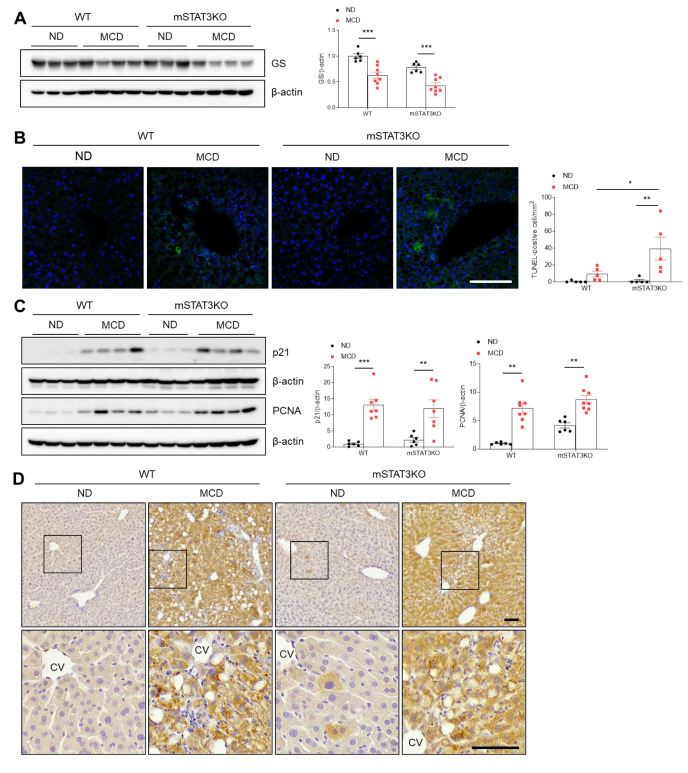
Effect of myeloid-specific STAT3 deletion on hepatic apoptosis in MCD diet-fed mice. (**A**) Western blot analysis and quantification of GS proteins in liver lysates (*n* = 6–8 mice per group). β-actin was used as loading control. (**B**) Representative TUNEL staining of liver sections and quantification of TUNEL-positive cells. Nuclei were stained with DAPI. Scale bar: 30 μm. (**C**) Western blot analysis and quantification of p21 and PCNA proteins in liver lysates (*n* = 6–8 mice per group). β-actin was used as loading control. (**D**) Immunohistochemistry for Ki67 in liver sections. Lower panels indicate magnified images of black line boxes. Scale bar: 60 μm. CV, central vein. Statistical significance was determined by using two-way ANOVA with Bonferroni’s post hoc testing. * *p* < 0.05, ** *p* < 0.001, and *** *p* < 0.0001.

**Figure 6 cells-14-01522-f006:**
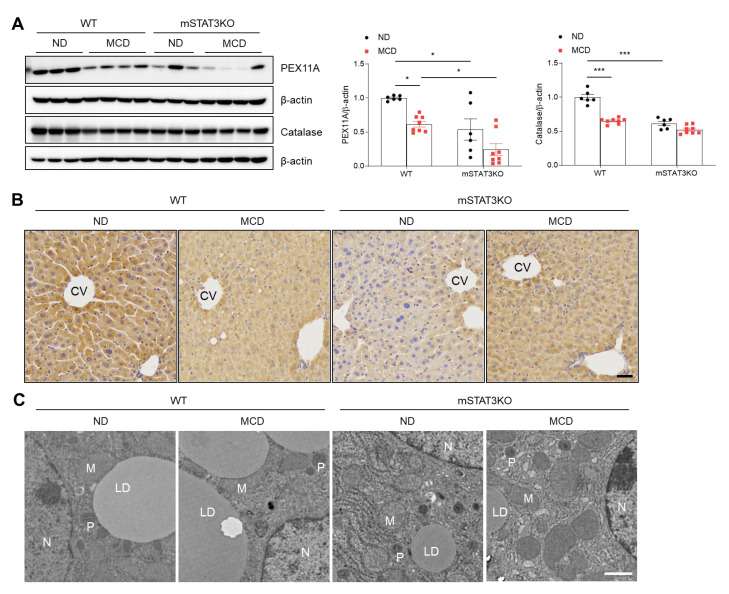
Effect of myeloid-specific STAT3 deletion on hepatic peroxisome activity in MCD diet-fed mice. (**A**) Western blot analysis and quantification of PEX11A and catalase proteins in liver lysates (*n* = 6–8 mice per group). β-actin was used as loading control. (**B**) Immunohistochemistry for PEX11A in liver sections. Scale bar: 30 μm. CV, central vein. (**C**) Transmission electron micrographs of hepatocytes. N, nucleus; M, mitochondria; P, peroxisome; LD, lipid droplet. Scale bar: 1 μm. Statistical significance was determined by using two-way ANOVA with Bonferroni’s post hoc testing. * *p* < 0.05, and *** *p* < 0.0001.

**Figure 7 cells-14-01522-f007:**
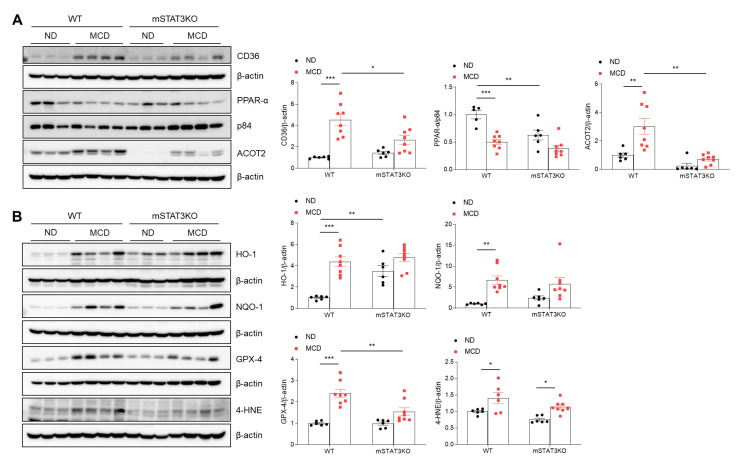
Effect of myeloid-specific STAT3 deletion on mitochondrial FA oxidation and oxidative stress in MCD diet-fed mice. (**A**) Western blot analysis and quantification of CD36, PPAR-α, and ACOT2 proteins in liver lysates (*n* = 5–8 mice per group). β-actin was used as loading control. (**B**) Western blot analysis and quantification of HO-1, NQO-1 GPX-4, and 4-HNE proteins in liver lysates (*n* = 6–8 mice per group). β-actin was used as loading control. Statistical significance was determined by using two-way ANOVA with Bonferroni’s post hoc testing. * *p* < 0.05, ** *p* < 0.001, and *** *p* < 0.0001.

**Figure 8 cells-14-01522-f008:**
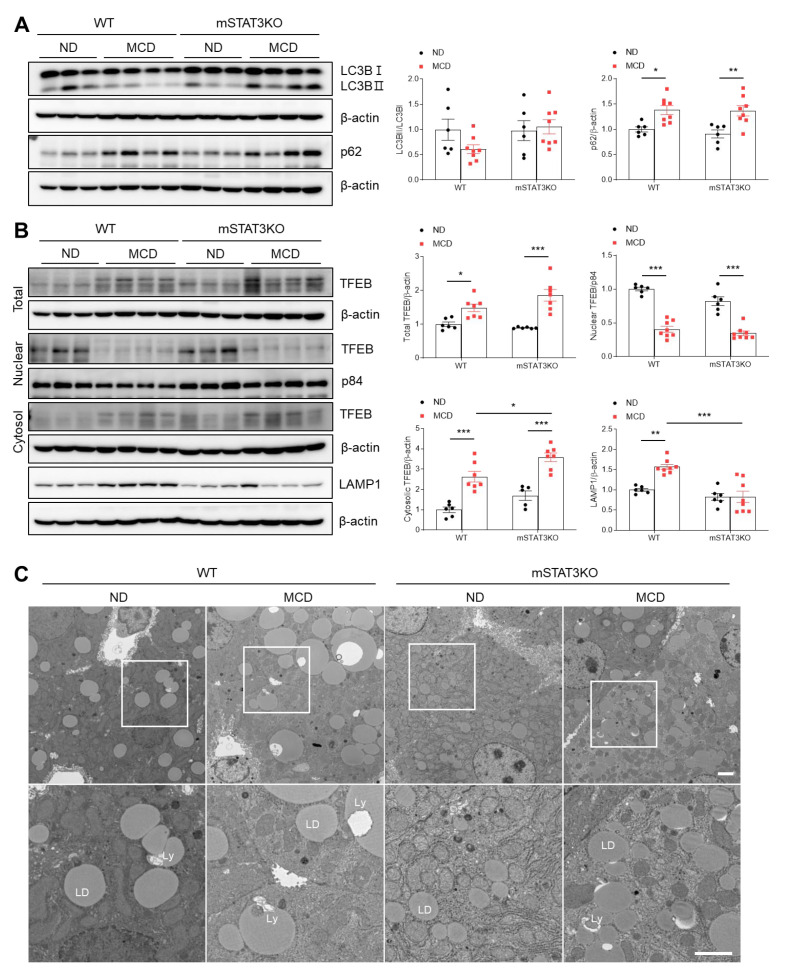
Effect of myeloid-specific STAT3 deletion on hepatic autophagic flux in MCD diet-fed mice. (**A**) Western blot analysis and quantification of LC3B and p62 proteins in liver lysates (*n* = 5–8 mice per group). β-actin was used as loading control. (**B**) Western blot analysis and quantification of TFEB and LAMP1 proteins in liver lysates (*n* = 5–8 mice per group). Nuclear p84 and total β-actin were used as loading controls. (**C**) Transmission electron micrographs of hepatocytes. Ly, lipolysosome; LD, lipid droplet. The bottom panels is an enlarged image of the white square box. Scale bar: 1 μm. Statistical significance was determined by using two-way ANOVA with Bonferroni’s post hoc testing. * *p* < 0.05, ** *p* < 0.001, and *** *p* < 0.0001.

## Data Availability

The data presented in this study are available in this manuscript.

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
