# Peer review of "Myeloid-Specific STAT3 Deletion Aggravates Liver Fibrosis in Mice Fed a Methionine- and Choline-Deficient Diet via Upregulation of Hepatocyte-Derived Lipocalin-2"

_cells, 2025, doi:10.3390/cells14191522_

Round 1

Reviewer 1 Report

Comments and Suggestions for Authors

Comments

  1. Since the MCD model induces significant weight loss (Figure 1A), how do you exclude the confounding effects of weight reduction and altered energy metabolism on inflammation and fibrosis?
  2. STAT3 has both pro- and anti-inflammatory roles depending on the cell type. Your results show that mSTAT3KO unexpectedly enhanced hepatocyte STAT3 activation. What could be the molecular mechanism behind this compensatory effect?
  3. Besides LCN2/MMP9, did you identify any other candidate factors associated with fibrosis that might be regulated by STAT3 activation in mSTAT3KO mice?
  4. Why do LCN2-deficient mice develop more severe steatosis under the MCD diet, whereas in your study LCN2 upregulation is associated with increased fibrosis? Does this suggest that LCN2 has dual roles depending on context?
  5. Have you considered applying multi-omics approaches (transcriptomics, proteomics, metabolomics) to comprehensively characterize the effects of mSTAT3KO on liver metabolism and inflammation?

Reviewer 2 Report

Comments and Suggestions for Authors

1.The manuscript would benefit from a more detailed explanation of the proposed mechanism by which myeloid-specific STAT3 deficiency may trigger a compensatory STAT3/LCN2 axis in hepatocytes, supported by relevant literature to strengthen the study’s rationale.

2.It is recommended that the figures clearly indicate each experimental group (WT vs. mSTAT3KO), treatment duration, and statistical significance, and that the figure legends specify the detection methods used.

3.The discussion section could be enhanced by addressing the potential clinical implications of myeloid-specific STAT3 deficiency in MASH and its relevance for anti-inflammatory or anti-fibrotic therapeutic strategies.

4.If the experiment only involves 4 weeks of MCD feeding, it is recommended to discuss whether this duration is sufficient to induce pronounced fibrosis and whether extending the feeding period for dynamic observation should be considered.

5.Consider analyzing whether downstream signaling pathways, such as JAK/STAT or NF-κB, are activated to strengthen the mechanistic credibility of the study.

6.The methods section should clearly specify the number of animals per group, the number of experimental replicates, and the statistical methods used to ensure the reliability of the results. For multiple comparisons, it is recommended to indicate the correction method employed (e.g., Bonferroni or FDR).

7.It is recommended that the discussion compare the results with existing studies on myeloid STAT3 in NASH/MASH models to highlight the novel findings and scientific significance of the current study.      
